:ᴏ̇: PLOS | ONE

# Persistent frequent emergency department users with chronic conditions: A population-based cohort study

Yohann Moanahere Chiu[1,2]*, Alain Vanasse[1,2], Josiane Courteau[1], Maud-Christine Chouinard[3], Marie-France Dubois[4], Nicole Dubuc[5], Nicolas Elazhary[1], Isabelle Dufour[1,5], Catherine Hudon[1,2]

1 Département de médecine de famille et de médecine d'urgence, Faculté de médecine et des sciences de la santé, Université de Sherbrooke, Sherbrooke, Quebec, Canada, 2 Centre de recherche du Centre hospitalier universitaire de Sherbrooke, Faculté de médecine et des sciences de la santé, Université de Sherbrooke, Sherbrooke, Quebec, Canada, 3 Département des sciences de la santé, Université du Québec à Chicoutimi, Chicoutimi, Quebec, Canada, 4 Département des sciences de la santé communautaire, Faculté de médecine et des sciences de la santé, Université de Sherbrooke, Sherbrooke, Quebec, Canada, 5 École des sciences infirmières, Université de Sherbrooke, Sherbrooke, Quebec, Canada

* yohann.chiu@usherbrooke.ca

**Data Availability Statement:** Our research team is bound by legal reasons to not divulge any part of the data. The Commission de l'accès à l'information du Québec (CAI) is the provincial

## Abstract

### Background

Frequent emergency department users are patients cumulating at least four visits per year. Few studies have focused on persistent frequent users, who maintain their frequent user status for multiple consecutive years. This study targets an adult population with chronic conditions, and its aims are: 1) to estimate the prevalence of persistent frequent ED use; 2) to identify factors associated with persistent frequent ED use (frequent use for three consecutive years) and compare their importance with those associated with occasional frequent ED use (frequent use during the year following the index date); and 3) to compare characteristics of "persistent frequent users" to "occasional frequent users" and to "users other than persistent frequent users".

### Methods

This is a retrospective cohort study using Quebec administrative databases. All adult patients who visited the emergency department in 2012, diagnosed with chronic conditions, and living in non-remote areas were included. Patients who died in the three years following their index date were excluded. The main outcome was persistent frequent use ($\geq$4 visits per year during three consecutive years). Potential predictors included sociodemographic characteristics, physical and mental comorbidities, and prior healthcare utilization. Odds ratios were computed using multivariable logistic regression.

### Results

Out of 297,182 patients who visited ED at least once in 2012, 3,357 (1.10%) were persistent frequent users. Their main characteristics included poor socioeconomic status, mental and

organisation that reviews research projects and allows researchers to access health databases. It is also responsible for ensuring their privacy as those databases contain sensitive patient information and it does not legally allow for making any part of them public. Therefore, we are not able to make any part of our data publicly available. Researchers interested in having access to databases used in this study (e.g. MED-ECHO, administrative and physician reimbursement registers) can submit a request to the Research data access point of the Institut de la statistique du Québec/CAI (https://www.stat.gouv.qc.ca/research/#/accueil).

**Funding:** This study was supported by the Fonds de recherche du Québec—Santé, the Quebec SPOR SUPPORT Unit, and the Centre de recherche du Centre hospitalier de l'université de Sherbrooke.

**Competing interests:** Dr Alain Vanasse has received grants for unrelated research from AstraZeneca Canada Inc. This does not alter our adherence to PLOS ONE policies on sharing data and materials. The authors declare no other competing interests.

physical comorbidity, and substance abuse. Those characteristics were also present for occasional frequent users, although with higher percentages for the persistent user group. The number of previous visits to the emergency department was the most important factor in the regression model. The occasional frequent users' attrition rate was higher between the first and second year of follow-up than between the second and third year.

## Conclusions

Persistent frequent users are a subpopulation of frequent users with whom they share characteristics, such as physical and mental comorbidities, though the former are poorer and younger. More research is needed in order to better understand what factors can contribute to persistent frequent use.

## Introduction

Frequent emergency department (ED) users constitute a small number of ED users, but account for a disproportionately large number of total ED visits [1]. Definition of frequent users varies, though the most common definitions include having more than three or four visits during a 12-month period [2, 3]. As each type of population has its own characteristics, those definitions are context dependent and therefore may vary. For instance, timely palliative care reduces number of ED visits near the end of life because of specific care given to patients [4, 5] while patients with asthma may require more ED visits, as they are more prone to exacerbations. [6, 7]. Even though, ED visits are not necessarily evitable, for instance regarding older adults [8]. Frequent ED users often receive non-optimal and fragmented care in EDs [9], due to their complex healthcare needs. Furthermore, they have higher hospital admissions and outpatient visits along with higher mortality rate [3]. In addition to higher healthcare costs, health outcomes associated with frequent ED use are non-optimal, in contrast to timely interventions from more appropriate health resources, such as in primary care [10]. A significant proportion of frequent ED users are patients with vulnerability factors, such as poor mental health [11], socio-economic precarity [12], or chronic conditions [1, 13, 14].

Among frequent users, a subgroup of persistent users keeps on visiting EDs frequently over a multiple-year period [1, 15–18]. Definitions of persistent frequent ED use vary in the number of visits per year (from more than three visits to more than five visits per year) and in the considered period (from a two-year period to a five-year period). Although their prevalence ranges from 1 to 20%, they can account for more than 60% of the total visit volume [2]. Factors associated with persistent frequent use are physical disorders, mental health disorders, substance abuse, previous number of ED visits, and being a frequent ED user the previous year [2]. Many studies have examined frequent ED use, but few have explored persistent frequent ED use. Besides, few studies have explored persistent frequent use considering chronic conditions.

The study targets an adult population with chronic conditions, and its aims are 1) to estimate the prevalence of persistent frequent ED use; 2) to identify factors associated with persistent frequent ED use (frequent use for three consecutive years) and compare their importance with those associated with occasional frequent ED use (frequent use during the year following the index date); and 3) to compare characteristics of "persistent frequent users" to "occasional frequent users" and to "users other than persistent frequent users" (which include non-frequent users and occasional frequent users).

## Materials and methods

This research was completed in accordance with the TRIPOD guidelines (see the Table in S1 Table) [19].

### Design and data sources

The provincial health insurance board of the Quebec Province (*Régie de l'assurance maladie du Québec* or RAMQ) is the provincial organism in charge of universal healthcare services for all of Quebec residents. We conducted a population-based retrospective cohort study using its health databases:

- The patient demographic register, which contains information about the sex, date of birth, date of death, and place of residence of the patient;

- The physician reimbursement claim register, which contains information about medical services provided by a fee-for-service physician in Quebec: date of service, place of service (emergency, medical clinic, etc.), physician specialty, diagnosis (International Classification of Diseases, ninth revision or ICD-9), and the medical act procedure performed by the physician;

- The hospital register (MED-ECHO), which contains information about the reasons for hospitalization (main diagnosis and up to 25 secondary diagnoses coded in ICD-10), dates of admission and release from hospital, and all medical procedures performed during the hospitalization.

Patient information from those databases were linked using unique encrypted identifiers.

### Study population

The study population included all adults ($\geq$18 years old) living in the province of Quebec, with at least one ED visit during the inclusion period, i.e. between the 1st of January 2012 and the 31st of December 2012, and diagnosed with one or more chronic conditions. In this study, we considered the Canadian Institute for Health Information definitions for ambulatory care sensitive conditions (see S1 and S2 Tables) [20]: asthma, chronic obstructive pulmonary disease or COPD, congestive heart failure or CHF, coronary heart disease or CHD, diabetes, epilepsy, and high blood pressure or HBP. Diagnoses were considered during a hospitalization or during two physician visits in the two-year period before the index date. The index date was randomly assigned as one ED visit among all ED visits occurring during the inclusion period [21].

There were two exclusion criteria (Fig 1). First, ED use can be different between urban and remote areas since remote residents tend to use it as an alternative to walk-in clinics as there are fewer primary care alternatives [22, 23]. Thus, patients living in municipalities with fewer than 10,000 inhabitants with weak or no metropolitan influence zone (remote areas, i.e. the percentage of resident employed labour force who commute to work in urban areas is less than 5%) were excluded from the study population. However, patients living in municipalities with fewer than 10,000 inhabitants with high or moderate metropolitan influence were included. Secondly, patients who died during the three years after their index date were excluded as they tend to require specialized healthcare (i.e. patients at the end of life [24, 25]). Besides, that last exclusion helps reduce immortal time bias [26]. Each patient had a follow-up time of three years.

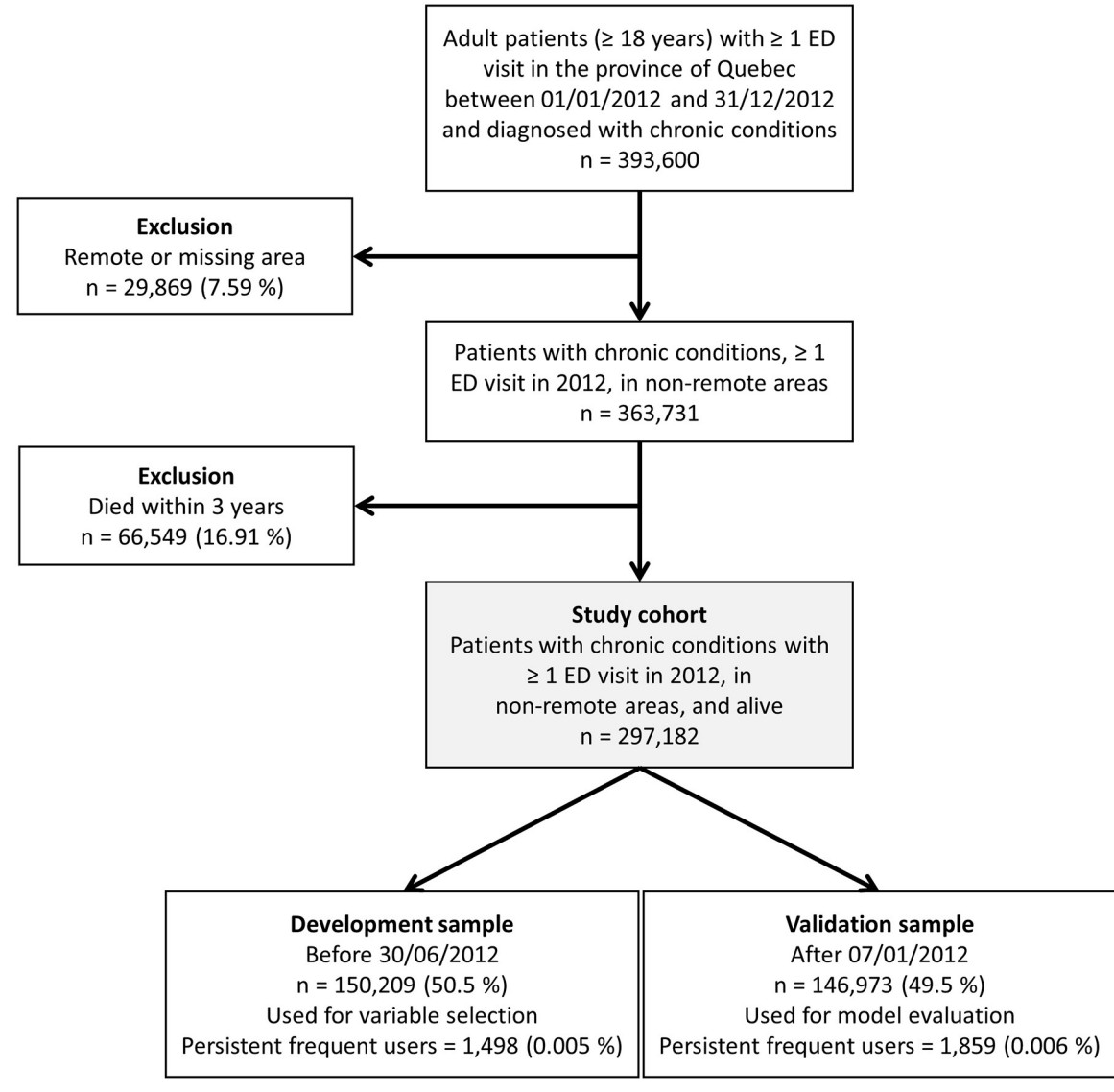

**Fig 1. Flowchart for the cohort selection.**

## Outcome and independent variables

The outcome was defined as being a persistent frequent user (binary variable, yes/no). In this study, frequent ED use was defined as having four or more ED visits in the year following their index date, while persistent frequent ED use was defined as frequent ED use during three consecutive years. To emphasize the difference between persistent frequent ED use and frequent ED use, we define a frequent ED user as "occasional frequent ED user". Occasional frequent ED users and persistent frequent ED users are two mutually exclusive categories, while "users other than persistent frequent users" exclude persistent frequent users but include occasional frequent users.

Independent variables considered were: sex, age category (18-34, 35-54, 55-64, 65-74, 75-84, and $\geq$85), the type of residential area (categorical variable: metropolitan: $\geq$100,000 inhabitants; small town: 10,000–100,000 inhabitants; rural: <10,000 inhabitants with high or moderate

metropolitan influence), material and social deprivation indices [27], public prescription drug insurance plan status (PPDIP, see below for the different statuses), diagnosis for each chronic condition (yes/no), diagnosis of depression or psychosis (yes/no), diagnosis of substance or drug abuse (yes/no), diagnosis of dementia (yes/no), having at least one hospitalization episode in the two years before the index date (yes/no), the number of ED visits during the year before the index date ($\leq$1, 2, 3, 4, $\geq$5), and the combined comorbidity index of Charlson and Elixhauser [28]. This index was modified to exclude the considered chronic conditions and was constructed using the following comorbidities: cardiac arrhythmia, any tumor without metastasis, peripheral vascular disorders, neurological disorders, cerebrovascular disease, renal disease, metastatic cancer, fluid and electrolyte disorders, liver disease, rheumatoid arthritis/collagen vascular disease, coagulopathy, weight loss, paralysis, and HIV/AIDS. The reported diagnoses in MED-ECHO (one diagnosis) or in the physician claims records (at least two diagnoses) were used to identify each condition during a two-year period before the index date.

Regarding PPDIP status, it is subdivided according to four different statuses: "not admissible to PPDIP" (individuals with a private insurance plan), "admissible to PPDIP and age $\geq$65 years with guaranteed income supplement" (GIS), "admissible to PPDIP and being a recipient of last-resort financial assistance" (LRFA), or "regular recipient of PPDIP".

## Statistical analysis

First, we reported the prevalence of persistent frequent ED use and the associated 99.9% confidence interval. Second, we used multivariable logistic regression to identify characteristics associated with persistent frequent ED use, since the outcome is a binary variable ("persistent frequent ED users" versus "users other than persistent frequent ED users"), and those associated with occasional frequent ED use ("occasional frequent ED users" versus "users other than occasional frequent ED users and persistent frequent ED users"). We reported odds ratios and associated 99.9% confidence intervals. Furthermore, given the small prevalence of the outcome, we used Firth's correction for logistic regression to reduce potential bias in the parameter estimations [29]. The models controlled for sex and age category. For others independent variables, automatic variable selection was implemented using backward selection, which consists in starting with a full model containing all variables, then deleting one at a time based on its statistical significance (Wald test) [30]. We used a split-sample approach in our models: we defined development and validation samples using a temporal split (at July 1st, 2012, Fig 1), as it is considered a stronger approach for developing and validating prognostic models [31]. Variable selection was performed on the development sample (first 50% of the cohort, n = 150,209), while odds ratios were obtained on the validation sample (remaining 50%, n = 146,973). Third, to compare characteristics between persistent frequent users and occasional frequent users or users other than persistent frequent users, chi-square tests of independence were used. Given the large sample size, statistical significance level was set at $\alpha$ = 0.001. All analyses were performed with SAS version 9.4.

## Ethics approval

The research ethics board of the *Centre intégré universitaire de santé et de services sociaux de l'Estrie–Centre hospitalier universitaire de Sherbrooke* approved this study. All data used in this study were fully anonymized.

## Results

Out of 297,182 patients who met the eligibility criteria, 1.1% (confidence interval 1.0-1.2%) were considered persistent frequent users (Table 1). Out of the 17,981 frequent users who were

**Table 1. Number of frequent users each year and relative to the first year.** Prevalence for persistent frequent use is in italic.

| | First year | Second year | Third year |
|---|---|---|---|
| **Frequent users each year** | | | |
| Total (n) | 17,981 | 20,700 | 22,387 |
| Percentage relative to cohort | 6.0 | 7.0 | 7.5 |
| **Frequent users who remain frequent users relative to first year** | | | |
| Total (n) | 17,981 | 6,132 | 3,357 |
| Percentage relative to cohort | 6.0 | 2.1 | *1.1* |
| Percentage relative to first year frequent users | 100 | 34.1 | 18.7 |

followed during their first year, 6,132 were still frequent users after two years and 3,357 after three years. Those users represented respectively 34.1% and 18.7% of the frequent users. The latter were thus characterized as persistent frequent users, as they maintained their status over three consecutive years. Furthermore, they used 9.5%, 9.2% and 8.9% of total ED visits during their three years of follow-up (Fig 2). Their number of ED visits per year ranged from 4 to 59 (median of 7 visits).

Odds ratios and model fit criteria for frequent use are presented in Table 2, based on the validation sample (n = 146,973). Two models are presented: one for occasional frequent use and one for persistent frequent use, both compared to the entire validation sample. Significant associated variables selected in the development sample (n = 150,209) were age, PPDIP admissibility, presence of COPD, CHD and diabetes, number of previous ED visits, comorbidity index, and diagnosis of depression or drug abuse. Besides those variables, occasional frequent ED use was associated with presence of asthma, coronary heart disease, and diabetes, social and material deprivation indices, and type of residential area. Being a recipient of LRFA and the previous number of ED visits were associated with the two largest odds

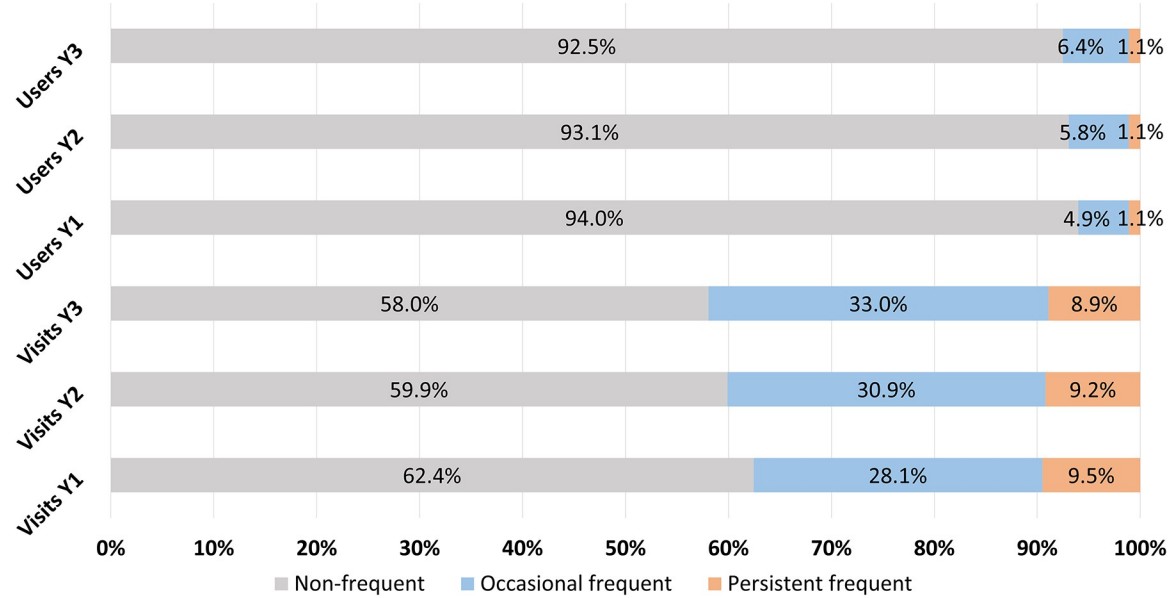

**Fig 2. Percentages of non-frequent (gray), occasional frequent (blue), and persistent frequent (orange) users relative to a) the number of ED visits (three bottom rows) and b) the number of ED users (three top rows) for each year of follow-up (Y1, Y2 and Y3).**

**Table 2. Odds ratios (99.9% confidence interval) and model fit criteria from multiple logistic regression for occasional frequent users and persistent frequent users, based on the validation sample (n = 146,973).**

| Variable | Occasional frequent use n = 10,510 | Persistent frequent use n = 1,859 |
|---|---|---|
| **Sex** | | |
| Male | Reference | |
| Female | 1.02 (0.94–1.09) | 1.14 (0.96–1.36) |
| **Age category** | | |
| 18–34 | 1.42 (1.21–1.67) | 1.57 (1.11–2.20) |
| 35–54 | 1.11 (0.98–1.25) | 1.20 (0.93–1.56) |
| 55–64 | Reference | |
| 65–74 | 0.97 (0.85–1.11) | 0.96 (0.69–1.33) |
| 75–84 | 1.10 (0.96–1.27) | 1.04 (0.74–1.46) |
| ≥85 | 1.14 (0.96–1.36) | 0.76 (0.49–1.17) |
| **PPDIP admissibility** | | |
| Regular | 1.11 (0.99–1.24) | 1.09 (0.82–1.46) |
| ≥65 years with GIS | 1.36 (1.18–1.56) | 1.65 (1.17–2.34) |
| Not admissible | Reference | |
| Recipients of LRFA | 1.76 (1.55–2) | 2.73 (2.09–3.59) |
| **Asthma** | 1.29 (1.16–1.43) | - |
| **Chronic obstructive pulmonary disease** | 1.46 (1.33–1.59) | 1.82 (1.52–2.17) |
| **Coronary heart disease** | 1.11 (1.02–1.21) | 1.17 (0.97–1.41) |
| **Diabetes** | 1.2 (1.11–1.29) | 1.37 (1.16–1.62) |
| **Number of previous ED visits** | | |
| ≤1 | Reference | |
| 2 | 2.53 (2.28–2.79) | 3.75 (2.82–4.98) |
| 3 | 3.78 (3.36–4.26) | 7.26 (5.41–9.69) |
| 4 | 5.63 (4.86–6.49) | 12.58 (9.20–17.09) |
| ≥5 | 12.47 (11.11–14) | 37.07 (29.33–47.07) |
| **Comorbidity index** | | |
| 0 | Reference | |
| 1–2 | 1.32 (1.21–1.44) | 1.40 (1.14–1.73) |
| 3–4 | 1.48 (1.31–1.66) | 1.38 (1.06–1.79) |
| ≥5 | 1.39 (1.24–1.56) | 1.28 (0.99–1.64) |
| **Social deprivation** | | |
| 1 | Reference | |
| 2 | 1.00 (0.87–1.14) | - |
| 3 | 1.07 (0.94–1.22) | - |
| 4 | 1.17 (1.03–1.33) | - |
| 5 | 1.23 (1.08–1.4) | - |
| **Material deprivation** | | |
| 1 | Reference | |
| 2 | 1.04 (0.90–1.2) | - |
| 3 | 1.10 (0.96–1.26) | - |
| 4 | 1.14 (1.00–1.31) | - |
| 5 | 1.16 (1.02–1.33) | - |
| **Residential area** | | |
| Metropolitan | Reference | |
| Small town | 1.16 (1.05–1.28) | - |
| Rural | 1.29 (1.16–1.42) | - |

*(Continued)*

**Table 2.** (Continued)

| Variable | Occasional frequent use n = 10,510 | Persistent frequent use n = 1,859 |
|---|---|---|
| Depression | 1.30 (1.18–1.43) | 1.34 (1.10–1.62) |
| Drug abuse | 1.55 (1.29–1.85) | 1.60 (1.20–2.13) |
| Area under the curve | 0.76 | 0.89 |
| R² | 7% | 28% |
| BIC | 65,093 | 14,885 |

-: the variable was not selected during the variable selection process.

ratios for persistent frequent ED use (respectively 2.8 and 39.1 for at 5 ED visits or more). We also evaluated our models on the full sample (without splitting) and we obtained similar results.

There were slightly more women in the persistent frequent user group than in the occasional frequent user group (59% and 56% respectively, Table 3). Persistent frequent users were also younger than occasional frequent users (higher proportion in the 35-54 category and lower proportions especially in the ≥85 category). Furthermore, there was a larger proportion of recipients of LRFA for persistent frequent users (31% versus 15% for occasional frequent users) and a reduced proportion of other PPDIP statuses. This indicates low socioeconomic standing, which relates to the distribution of those users in the higher material and social deprivation indices. However, repartition in the residential areas was similar. Except for high blood pressure, persistent frequent users had higher chronic condition prevalence than the occasional frequent users had. Persistent frequent users also had higher comorbidity indices. Furthermore, an important proportion of persistent frequent users were already frequent users in the year before their index dates since more than 60% had four or more previous ED visits. In comparison, this percentage was 28% for occasional frequent users. Drug and alcohol abuse, depression and psychoses were also more prevalent in the persistent frequent user group, whereas dementia was slightly less prevalent. Overall, the differences mentioned in this paragraph were also noticeable between users other than persistent frequent users, and occasional frequent users and persistent frequent users. For instance, chronic obstructive pulmonary disease prevalence increased from 14% to 25% and 37% for each type of users, respectively.

## Discussion

To the best of our knowledge, this work is the first to focus on persistent frequent ED users with chronic conditions. In this study, persistent frequent users represented 1.1% of the cohort, adding up to 9% of total ED visits each year. This prevalence is consistent with other studies about persistent frequent users in children and nonelderly adults [32–34]. Those studies used the same definition (≥4 ED visits during three consecutive years). Hudon *et al.* (2017) reported a higher prevalence of 2.6%, but their threshold for defining persistent frequent use was lower (≥3 ED visits during three consecutive years) and they focused on a diabetic population [35].

The variables associated with persistent frequent use found in this study were all reported by previous studies [32, 33, 35, 36], while some studied both occasional frequent use and persistent frequent use [32, 33, 36]. In those same studies, depending on the studied population, some specific diagnoses were also associated with persistent frequent use, such as COPD diagnosis. Besides, a few studies mentioned race [32, 33] and deprivation indices [32, 35, 36] as significant, though those latter were not necessarily the same indices as ours. Race was not

**Table 3. Descriptive statistics for the cohort, users other than persistent frequent users, occasional frequent users, and persistent frequent users.**

| Variable | Total | Users other than persistent frequent users | Occasional frequent users | Persistent frequent users |
|---|---|---|---|---|
| **Total** | 297,182 (100) | 293,825 (100) | 14,624 | 3,357 (100) |
| **Sex** | | | | a, b |
| Female | 158,881 (53.5) | 156,896 (53.4) | 8,171 (55.9) | 1,985 (59.1) |
| Male | 138,301 (46.5) | 136,929 (46.6) | 6,453 (44.1) | 1,372 (40.9) |
| **Age** | | | | a, b |
| 18–34 | 17,640 (5.9) | 17,301 (5.9) | 1,155 (7.9) | 339 (10.1) |
| 35–54 | 58,072 (19.5) | 57,224 (19.5) | 2,806 (19.2) | 848 (25.3) |
| 55–64 | 64,372 (21.7) | 63,757 (21.7) | 2,640 (18.1) | 615 (18.3) |
| 65–74 | 73,591 (24.8) | 72,904 (24.8) | 3,305 (22.6) | 687 (20.5) |
| 75–84 | 61,041 (20.5) | 60,370 (20.5) | 3,285 (22.5) | 671 (20.0) |
| ≥85 | 22,466 (7.6) | 22,269 (7.6) | 1,433 (9.8.) | 197 (5.9) |
| **PPDIP admissibility** | | | | a, b |
| Regular | 109,034 (36.7) | 108,216 (36.8) | 4,677 (32.0) | 818 (24.4) |
| ≥65 years with GIS | 77,638 (26.1) | 76,632 (26.1) | 4,679 (32.0) | 1,006 (30.0) |
| Not admissible | 85,621 (28.8) | 85,116 (29.0) | 3,024 (20.7) | 505 (15.0) |
| Recipients of LRFA | 24,889 (8.4) | 23,861 (8.1) | 2,244 (15.3) | 1,028 (30.6) |
| **Residential area** | | | | b |
| Metropolitan | 196,791 (66.2) | 194,737 (66.3) | 9,083 (62.1) | 2,054 (61.2) |
| Small town | 45,605 (15.3) | 44,988 (15.3) | 2,552 (17.5) | 617 (18.4) |
| Rural | 54,786 (18.4) | 54,100 (18.4) | 2,989 (20.4) | 686 (20.4) |
| **Coronary heart disease** | 75,564 (25.4) | 74,347 (25.3) | 4,714 (32.2) | 1,217 (36.3) a, b |
| **Asthma** | 34,291 (11.5) | 33,465 (11.4) | 2,369 (16.2) | 826 (24.6) a, b |
| **Chronic obstructive pulmonary disease** | 43,307 (14.6) | 42,051 (14.3) | 3,619 (24.7) | 1,256 (37.4) a, b |
| **Congestive heart failure** | 17,505 (5.9) | 17,083 (5.8) | 1,592 (10.9) | 422 (12.6) a, b |
| **Diabetes** | 96,983 (32.6) | 95,620 (32.5) | 5,270 (36.0) | 1,363 (40.6) a, b |
| **Epilepsy** | 8,600 (2.9) | 8,361 (2.8) | 709 (4.8) | 239 (7.1) a, b |
| **High blood pressure** | 163,132 (54.9) | 161,234 (54.9) | 8,314 (56.9) | 1,898 (56.5) |
| **Number of ED visits (1 year before the index date)** | | | | a, b |
| ≤1 | 221,524 (74.5) | 221,021 (75.2) | 6,124 (41.9) | 503 (15.0) |
| 2 | 36,457 (12.3) | 36,079 (12.3) | 2,542 (17.4) | 378 (11.3) |
| 3 | 17,830 (6.0) | 17,414 (5.9) | 1,876 (12.8) | 416 (12.4) |
| 4 | 8,931 (3.0) | 8,578 (2.9) | 1,248 (8.5) | 353 (10.5) |
| ≥5 | 12,440 (4.2) | 10,733 (3.7) | 2,834 (19.4) | 1,707 (50.8) |
| **Previous hospitalization in the last two years** | 135,257 (45.5) | 132,682 (45.2) | 9,293 (63.5) | 2,575 (76.7) a, b |
| **Material deprivation** | | | | a, b |
| Missing | 21,969 (7.4) | 21,654 (7.4) | 1,313 (9.0) | 315 (9.4) |
| 1 | 42,397 (14.3) | 42,088 (14.3) | 1,620 (11.1) | 309 (9.2) |
| 2 | 51,146 (17.2) | 50,716 (17.3) | 2,219 (15.2) | 430 (12.8) |
| 3 | 55,375 (18.6) | 54,797 (18.6) | 2,619 (17.9) | 578 (17.2) |
| 4 | 61,847 (20.8) | 61,107 (20.8) | 3,215 (22.0) | 740 (22.0) |

*(Continued)*

**Table 3.** (Continued)

| Variable | Total | Users other than persistent frequent users | Occasional frequent users | Persistent frequent users |
|---|---|---|---|---|
| 5 | 64,448 (21.7) | 63,463 (21.6) | 3,638 (24.9) | 985 (29.3) |
| **Social deprivation** | | | | [a, b] |
| Missing | 21,969 (7.4) | 21,654 (7.4) | 1,313 (9.0) | 315 (9.4) |
| 1 | 46,570 (15.7) | 46,181 (15.7) | 1,883 (12.9) | 389 (11.6) |
| 2 | 49,885 (16.8) | 49,483 (16.8) | 2,123 (14.5) | 402 (12.0) |
| 3 | 55,220 (18.6) | 54,632 (18.6) | 2,555 (17.5) | 588 (17.5) |
| 4 | 58,378 (19.6) | 57,723 (19.6) | 2,928 (20.0) | 655 (19.5) |
| 5 | 65,160 (21.9) | 64,152 (21.8) | 3,822 (26.1) | 1,008 (30.0) |
| **Comorbidity index** | | | | [a, b] |
| 0 | 179,430 (60.4) | 178,329 (60.7) | 6,424 (43.9) | 1,101 (32.8) |
| 1–2 | 66,097 (22.2) | 65,132 (22.2) | 3,868 (26.4) | 965 (28.7) |
| 3–4 | 24,240 (8.2) | 23,673 (8.1) | 2,007 (13.7) | 567 (16.9) |
| ≥5 | 27,415 (9.2) | 26,691 (9.1) | 2,325 (15.9) | 724 (21.6) |
| **Alcohol abuse** | 8,644 (2.9) | 8,191 (2.8) | 1,023 (7.0) | 453 (13.5) [a, b] |
| **Depression** | 36,601 (12.3) | 35,473 (12.1) | 3,098 (21.2) | 1,128 (33.6) [a, b] |
| **Drug abuse** | 5,817 (2.0) | 5,323 (1.8) | 885 (6.1) | 494 (14.7) [a, b] |
| **Psychoses** | 11,498 (3.9) | 11,013 (3.7) | 1,266 (8.7) | 485 (14.4) [a, b] |
| **Dementia** | 12,189 (4.1) | 11,960 (4.1) | 1,023 (7.0) | 229 (6.8) [b] |

Percentages in parentheses are relative to the column total.

[a] chi-square test of independence significant between persistent and occasional frequent users.

[b] chi-square test of independence significant between persistent frequent users and other users than persistent frequent users.

available in our databases and deprivation indices were not significant in our analyses of persistent frequent use. Andren and Rosenqvist (1987) mentioned that patient's loneliness and good rating of how they have been received at the ED increased the risk of returning to the ED in the next year in a two-year follow-up study [15]. Those variables had been collected during interviews and were not available in our databases. Thus, it would be relevant to include self-reported variables in a future work, complementary to administrative data.

Understanding reasons that may lead occasional frequent use to persistent frequent use is not trivial, as one type was not so different from the other one. In our study, they both were patients with a high comorbidity burden, diagnosed with depression and drug abuse, and with a history of ED visits. Regarding occasional frequent users with chronic conditions, Hudon *et al.* (2019) highlighted those same characteristics [21]. However, there were differences in our results. For instance, the social deprivation index, diagnosis of diabetes, or the type of residential area were not included in the persistent frequent use model, although they were in the occasional frequent use one. Furthermore, persistent frequent users were younger and had a heavier ED history (60% had more than four ED visits while this proportion was 27% for occasional frequent users). Finally, odds ratios for PPDIP status (being a recipient of LRFA) and number of previous ED visits were larger for persistent frequent use than for occasional frequent use, while other odds ratios were comparable in the two cases. We are not aware of other studies comparing those two variables between persistent frequent and occasional frequent users, but some reported Medicaid, an American federal insurance for patients with low-income amongst others [37], as associated with persistent frequent use [32, 38].

Previous use of ED turned out to be the most important associated factor for all the models, though its impact was stronger in the case of persistent frequent use (larger odds ratios) than in the case of occasional frequent use. The importance of previous ED use has been stated in previous studies, in the cases of occasional frequent use and persistent frequent use [2, 6, 17, 39, 40]. In particular, when studied in the baseline year, it has been reported as the strongest predictor of persistent frequent ED use in the subsequent year [32]. This may be explained by the fact that many occasional frequent ED users will not keep on visiting ED frequently and most of them will not have as many previous ED visits as already established persistent frequent ED users. Thus, previous use of ED has greater importance when it comes to studying persistent frequent ED use.

Many of the persistent frequent users in this study were already frequent users in the year before their index dates. They may have been frequent users for an even longer time than studied here. More precisely, frequent use attrition (i.e. proportion of frequent users who do not maintain their status the following year) was higher after the first year, compared to the second year (Table 1). Relative to the first-year frequent users, 34.0% maintained their status over the next year and 18.7% over the next two years. Two other studies reported this rate of decline slowing after the first year, with similar rates [15, 18], though they did not focus on a population with chronic conditions. In particular, Mandelberg *et al.* (2000) found that frequent users in their first year had a probability of 37.9% to maintain their status for another consecutive year [18]. This probability increased to 56.1% after two years of frequent use and to 78.7% after five years of frequent use. This suggests a "core" group in persistent frequent users.

Targeting persistent frequent users for specific interventions (such as case management [10]) may be even more relevant than targeting occasional frequent users. However, identifying them may also be more challenging. Firstly, administrative and medical data usually do not include self-perceived variables. Secondly, the high imbalance of class (persistent frequent users represent 1.1% of our cohort) means that traditional statistical models may not perform well. Since the majority of cases are not persistent frequent users, most of the statistical models will mainly use information from those cases, resulting in a suboptimal use of the information. One study showed that when the class of interest amounts to 1% of the observations, logistic regression has limited power compared to more advanced techniques, such as random forests [41]. Some adaptations exist for imbalanced data, but none has been applied to frequent ED users yet [42]. For instance, artificially balanced datasets or cost-sensitive methods, which have been used in medical sciences [43, 44], could be of interest for increasing the classification performances.

## Limitations

We did not include self-perceived variables, such as physical pain or mental distress. Those variables were not available in our databases and are associated with occasional frequent use [45, 46], though they have not been studied with persistent frequent use. We also did not have access to financial status or education at the individual level. Proxy variables such as social and material deprivation and PPDIP were used instead. Lastly, our study investigated ED users with chronic conditions (defined as 1 hospitalization or 2 physician visits related to a chronic condition in the two year period before the index date, though we did not investigate the reasons for ED visits during the follow-up period), thus limiting its generalization to all ED users.

## Conclusions

This paper focuses on persistent frequent ED users with chronic conditions. It highlights the fact that those users, who are a special case of frequent ED users over three consecutive years

(1.1% of the total cohort), share similarities with occasional frequent users such as physical and mental comorbidities, though with higher rates. However, they are younger and poorer than occasional frequent users. Those characteristics would make them priority targets for specialized interventions. More studies are needed in order to fully characterize persistent frequent use and to understand what factors can transform occasional frequent use into persistent frequent use, especially using other databases than administrative ones or specialized statistical methods for imbalanced data.

## Supporting information

**S1 Table. TRIPOD checklist for reporting cohort studies.**
(DOCX)

**S2 Table. International classification of diseases for diagnoses used in this study.**
(DOCX)

## Acknowledgments

The authors would like to thank Tina Wey (PhD) for revising the text and two anonymous reviewers whom helped improve the quality of this paper.

## Author Contributions

**Conceptualization:** Yohann Moanahere Chiu, Alain Vanasse, Josiane Courteau, Maud-Christine Chouinard, Marie-France Dubois, Nicole Dubuc, Nicolas Elazhary, Catherine Hudon.

**Data curation:** Yohann Moanahere Chiu, Josiane Courteau.

**Formal analysis:** Yohann Moanahere Chiu, Alain Vanasse, Josiane Courteau, Marie-France Dubois, Catherine Hudon.

**Funding acquisition:** Alain Vanasse, Josiane Courteau, Maud-Christine Chouinard, Marie-France Dubois, Nicole Dubuc, Nicolas Elazhary, Catherine Hudon.

**Investigation:** Yohann Moanahere Chiu, Alain Vanasse, Josiane Courteau, Maud-Christine Chouinard, Marie-France Dubois, Nicole Dubuc, Nicolas Elazhary, Catherine Hudon.

**Methodology:** Yohann Moanahere Chiu, Alain Vanasse, Josiane Courteau, Maud-Christine Chouinard, Marie-France Dubois, Nicole Dubuc, Nicolas Elazhary, Catherine Hudon.

**Project administration:** Alain Vanasse, Maud-Christine Chouinard, Marie-France Dubois, Nicole Dubuc, Nicolas Elazhary, Catherine Hudon.

**Resources:** Alain Vanasse, Nicole Dubuc, Catherine Hudon.

**Software:** Yohann Moanahere Chiu, Josiane Courteau, Isabelle Dufour.

**Supervision:** Alain Vanasse, Maud-Christine Chouinard, Marie-France Dubois, Nicole Dubuc, Nicolas Elazhary, Catherine Hudon.

**Validation:** Yohann Moanahere Chiu, Alain Vanasse, Josiane Courteau, Maud-Christine Chouinard, Marie-France Dubois, Nicole Dubuc, Nicolas Elazhary, Isabelle Dufour, Catherine Hudon.

**Visualization:** Yohann Moanahere Chiu, Alain Vanasse, Josiane Courteau, Catherine Hudon.

**Writing – original draft:** Yohann Moanahere Chiu, Catherine Hudon.

**Writing – review & editing:** Yohann Moanahere Chiu, Alain Vanasse, Josiane Courteau, Maud-Christine Chouinard, Marie-France Dubois, Nicole Dubuc, Nicolas Elazhary, Isabelle Dufour, Catherine Hudon.

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
