## [Decision Letter · Decision Letter 0]

7 Aug 2019

PONE-D-19-18906

Persistent frequent emergency department users with chronic ambulatory care sensitive conditions: a population-based cohort study

PLOS ONE

Dear Mr Chiu,

Thank you for submitting your manuscript to PLOS ONE. After careful consideration, we feel that it has merit but does not fully meet PLOS ONE’s publication criteria as it currently stands. Therefore, we invite you to submit a revised version of the manuscript that addresses the points raised during the review process.

The methodology of this study presents several shortcomings and there are relevant points in the manuscript that need further clarification:

The criteria for inclusion and exclusion of patients are questionable. Please, reconsider it according to the comments of the reviewers.Besides, I have serious doubts about the use of ACSCs in this study. The authors selected patients with “at least one ED visit during the inclusion period, i.e. between the 1st of January 2012 and the 31st of December 2012, and diagnosed with an ACSC “. It would seem logical to assume that all the ED visits were categorized into ACSC and non-ACSC; afterward, the patients with an ACSC visit would have been followed for a three year period. However, I am not sure that this was the case. There is not any mention in the manuscript of the diagnoses or reasons for visiting the ED, but antecedents of chronic diseases instead. The concept of ACSCs is referred to conditions for which the adequate use of primary care services could avoid admissions or ED visits. Consequently, it is not appropriate to regard as ACSC the visits due to other reasons in patients whose chronic pathologies are adequately controlled in primary care or elsewhere.In the statistical analyses it is not clear the reason for splitting the sample into training and validation ones.Also, the comparison of the results of only two groups (“persistent & frequent” vs. “frequent but not persistent”) seems insufficient. A better option will be to include also a third group of “no frequent users” .Follow all the valuable comments of the reviewers.

We would appreciate receiving your revised manuscript by Sep 21 2019 11:59PM. To enhance the reproducibility of your results, we recommend that if applicable you deposit your laboratory protocols in protocols.io, where a protocol can be assigned its own identifier (DOI) such that it can be cited independently in the future. For instructions see: http://journals.plos.org/plosone/s/submission-guidelines#loc-laboratory-protocols

We look forward to receiving your revised manuscript.

Kind regards,

Juan F. Orueta, MD, PhD

Academic Editor

PLOS ONE

Journal Requirements:

"The research ethics board of the CIUSSS Estrie - CHUS approved this study.".

i) Please amend your current ethics statement to include the full name of the ethics committee/institutional review board(s) that approved your specific study.

ii) Once you have amended this/these statement(s) in the Methods section of the manuscript, please add the same text to the “Ethics Statement” field of the submission form (via “Edit Submission”).

3. Thank you for submitting the above manuscript to PLOS ONE. We note you have stated the following in the data availability statement:

"The datasets generated and/or analysed during the current study are not publicly available due to individual privacy. The CAI (Commission d'accès à l'information duQuébec) does not authorize data sharing outside of the research team."

Please note that PLOS journals require authors to make all data underlying the findings described in their manuscript fully available without restriction, with rare exception. PLOS journals will not consider manuscripts for which authors will not share data because of personal interests, such as patents, commercial interests or potential future publications. Your current statement in not in line with our data availability policy, which can be found at the following link: https://journals.plos.org/plosone/s/data-availability

Please update your Data Availability Statement to indicate whether you will be able to make your data available at the time of acceptance and provide details of where the underlying dataset can be found. Please be aware that without such confirmation we won't be able to consider your manuscript for publication.   

Please also note that we only require you to provide the minimal dataset, to consist of the data set used to reach the conclusions drawn in the manuscript with related metadata and methods, and any additional data required to replicate the reported study findings in their entirety. We do not require that you submit the entire data set if only a portion of the data were used in the reported study.

Thank you for your attention to this request.

4. In ethics statement in the manuscript and in the online submission form, please provide additional information about the patient records used in your retrospective study. Specifically, please ensure that you have discussed whether all data were fully anonymized before you accessed them and/or whether the IRB or ethics committee waived the requirement for informed consent. If patients provided informed written consent to have data from their medical records used in research, please include this information."

Alain Vanasse has received grants for unrelated research from AstraZeneca Canada Inc. The authors declare no other competing interests.

We note that you received funding from a commercial source:AstraZeneca Canada Inc

6. We note that you have indicated that data from this study are available upon request. PLOS only allows data to be available upon request if there are legal or ethical restrictions on sharing data publicly. For information on unacceptable data access restrictions, please see http://journals.plos.org/plosone/s/data-availability#loc-unacceptable-data-access-restrictions.

7. Please amend either the abstract on the online submission form (via Edit Submission) or the abstract in the manuscript so that they are identical.

Reviewers' comments:

Reviewer's Responses to Questions

**Comments to the Author**

1. Is the manuscript technically sound, and do the data support the conclusions?

Reviewer #1: Partly

Reviewer #2: Yes

2. Has the statistical analysis been performed appropriately and rigorously? 

Reviewer #1: Yes

Reviewer #2: Yes

3. Have the authors made all data underlying the findings in their manuscript fully available?

Reviewer #1: No

Reviewer #2: Yes

4. Is the manuscript presented in an intelligible fashion and written in standard English?

Reviewer #1: Yes

Reviewer #2: Yes

5. Review Comments to the Author

Reviewer #1: Thank you for the opportunity to read this manuscript, which concerns itself with a) estimating the prevalence of persistent frequent ED use, 2) identifying factors associated with this use and 3) comparing characteristics of frequent and non-frequent users. While I have offered comments to strengthen the paper should these authors resubmit to PLOS-One or elsewhere, I am not sure that it can make a contribution to this literature for reasons I detail in my comments. Ultimately, the fact that the study is described as a population based cohort study but the outcome variable was operationalized for only a subset of cases (and the fact that this is not clear in the manuscript) detract from the paper and preclude me from being able to recommend it for publication.

I offer comments by section.

Introduction

My initial comments were relative to the number of and appropriateness of the references in the introduction section. I would like to have seen more references (and potentially more explication) for the first set of assertions in the paper (e.g. the entire first paragraph). For example, even the second sentence (“Definition of frequent users varies depending on the context, though the most common definitions include having more than three or four visits during a 12-month period”) only includes one citation, the pertinence of which is not immediately appreciable. The premises in this paragraph form the bulk of their argument and feel insufficiently motivated (referenced) and under also under-nuanced. Is this true for cancer patients, patients with cancer related pain, the frail elderly, etc.? I am, in general, against language that suggests that patients “over-use” emergency care, when the situations of these patients’ lives is often such that it does represent the best care option for them at the time they choose it. Similarly, what is meant by ‘vulnerable’ patients on line 60? More detail and nuanced description of patterns of emergency care use are needed overall in the introduction. Although they have cited Billings (2013), it feels as through they have missed the subtleties of this piece.

As an other example, the exclusion from the cohort of patients who had died within 3 years of their index visit strengthened my opinion that the intro section could do a better job with the nuances of what constitutes ‘frequent and persistent use’ as in some cases (terminally ill patients) there may well be another definitions. I had also initially commented that looking deeper into this literature would also provide guidance for these investigators about whether it is in fact true that “no study exploring persistent frequent use with ACSC has been published.”

However – as I read further in the manuscript and identified the issue of ‘outcome cases’ vs ‘reference cases’, and realized that the analyses in the paper are actually relative to ‘persistent and frequent users’ vs ‘frequent users’, I felt that the intro should also be revised in light of this focus. The reader assumes that the study is about ‘high users’ vs ‘non high users’ when in in fact is it about ‘persistent high users’ vs ‘episodic high users’. The literature review provided does little to shed light on hypothesis that may relate to ‘those with frequent use’ vs ‘those with persistent frequent use’. In fact, it is a bit misleading to imply that this is a ‘population based cohort study’ when in fact less than 10% of the sample is utilized for the primary analytic aim of the study.

Materials & Methods Section

Consistent with my comment above – I felt that the method section should make it clearer early on what cases comprise the actual analytic sample in this study. The reader does not know until Table 3. Figure 1 should be similarly revised to reflect this.

Other comments:

Is the RAMQ datasource actually exhaustive? i.e. is it literally all adult patients in the province? Please detail, for American readers this is going to be a difficult concept to grasp. Does it contain all the variables used, or is it matched with (an)other data source? It wasn’t entirely clear to me if the RAMQ and the RAMQ administrative data are the same. If not, was it possible to match all cases in the cohort across the databases used? Cases lost (if any) during this process should be detailed in Fig. 1.

As cases were excluded for rural residential status with no/low metropolitan influence (but cases with rural status with high metropolitan influence were included), this variable should be more clearly defined.

The splitting of the testing and validation samples should be described in more detail, this is a somewhat unusual technique and deserves further explication. What is the rationale for this approach over others? Does it have any implications if there are temporal dependencies in the data?

Most importantly, and weaved throughout all my comments: In the section ‘Outcome and Dependent variables’, it should be made clearer that the binary outcome was ‘persistent & frequent users’ compared with the reference category ‘frequent but not persistent’. I was somewhat surprised when I reached the results section to see that this was the case. It is natural, given the language used and overall framing of this study, for the reader to assume that the reference category in analyses is going to be ‘non frequent (non persistent) user’ (e.g. the rest of the sample). This should also be addressed in figure 1, as the reader cannot tell (easily) from either the text or the figure what N was actually used in analyses (one does not see it until Table 3). It is difficult to tell if everyone who was not in the ‘persistent’ category were by default included in the other category? There must certainly have been cases for whom the index visit was the only visit? Is it fair to categorize them as ‘frequent but not persistent’? This is quite confusing until one reaches Table 3.

I was going to comment on the applicability of logistic regression with such a low frequency of cases on the outcome compared to the reference category, until I realized what was going on with the reference category (there is in fact a reasonable distribution of cases to controls in the analytic sample). Unless I am misunderstanding, these analyses really only include the approximately 13,000 cases categorized as either ‘persistent frequent’ or ‘frequent but not persistent’. This has to be made clearer throughout the manuscript. The rationale for not using the other (more obvious) reference category should be presented and as I said above, the introduction section should be made consistent with this approach.

However, the paper still requires more detail on the backward selection technique used – did it result in only sex and age category as control variables, as is implied? This is confusing as Table 2 presents the odds ratios for many predictors in the multivariable model.

Results

As the sample was split for training and validation it should be clarified in the Tables what sample(s) were used for which analyses. It is not stated what the N’s are that resulted from this process (size of each subsample), and as I highlight below, it remains somewhat unclear throughout the rest of the paper when the split samples were used, and what size the split samples were with respect to the total, frequent users, and persistent users. The fact that this study rests on approx. 13,000 cases is fine, but it should not be represented in the paper as being based on the full sample.

Reviewer #2: The manuscript by Chiu YM et al. analyze in an adult population characterized by an ambulatory care sensitive conditions (ACSC) the prevalence of and the factors eventually associated to the condition of persistent frequent users in the Emergency Department. The manuscript is clearly written, data are well collected and the conclusions are well supported by novel and interesting results.

My main methodological concern is related to some criteria used for the selection of the experimental sample. In particular previous studies (see in particular PLOS ONE |2016 Dec 14;11(12):e0165939. doi: 10.1371/journal.pone.0165939. eCollection 2016.) have shown that older patients present clinical and social characteristics related to the definition of “elderly frail frequent users”. This data do not seem to be confirmed in the present study. Is it possible that the exclusion of patients affected by dementia and of patients died in the three years following their index may have caused a bias in the interpretation of data considering that dementia is strictly correlated with the old age? The authors need to discuss this issue.

Furthermore the authors state that previous use of ED turned out to be the most important factor for all methods in encouraging the transition from occasional to persistent frequent users. My question is why does this happen? Which are the mechanisms linking the previous ED use to the transition in persistent frequent user?

Can the authors provide some data about the hospital admission of the occasional and persistent frequent user? It is not clear to the reviewer the data “Previous hospitalization” in Tab. 3, please clarify this issue and eventually discuss these data.

Page 15, line 235: What do the authors mean for “...heavier ED history.” ?

6. PLOS authors have the option to publish the peer review history of their article (what does this mean?). If published, this will include your full peer review and any attached files.

Reviewer #1: No

Reviewer #2: No

---

## [Author Response · Author response to Decision Letter 0]

25 Oct 2019

October 10th, 2019

PLOS ONE

Dear Dr Orueta,

We would like to thank the editors and reviewers for their comments on our manuscript, “Persistent frequent emergency department users with chronic ambulatory care sensitive conditions: a population-based cohort study”, which offered us the opportunity to considerably strengthen it. You will find attached the revised version. Moreover, you will find detailed responses to each of the comments in this letter.

We hope the revisions will be to your satisfaction and we look forward to hearing back from you.

Best regards,

Yohann M. Chiu, Ph.D.

Corresponding Author

Université de Sherbrooke

3001, 12e Avenue Nord

Sherbrooke, QC J1H 5N4

Phone: 819-346-1110, #70538

Email: yohann.chiu@usherbrooke.ca

 

Editor’s comments 

The criteria for inclusion and exclusion of patients are questionable. Please, reconsider it according to the comments of the reviewers.

Response: We have updated the text according to the comments of the reviewer (see below for detailed answers).

Besides, I have serious doubts about the use of ACSCs in this study. The authors selected patients with “at least one ED visit during the inclusion period, i.e. between the 1st of January 2012 and the 31st of December 2012, and diagnosed with an ACSC “. It would seem logical to assume that all the ED visits were categorized into ACSC and non-ACSC; afterward, the patients with an ACSC visit would have been followed for a three year period. However, I am not sure that this was the case. There is not any mention in the manuscript of the diagnoses or reasons for visiting the ED, but antecedents of chronic diseases instead. The concept of ACSCs is referred to conditions for which the adequate use of primary care services could avoid admissions or ED visits. Consequently, it is not appropriate to regard as ACSC the visits due to other reasons in patients whose chronic pathologies are adequately controlled in primary care or elsewhere.

Response: Thank you for pointing out this issue, we added it to the limits. ACSCs were an inclusion criterion for the cohort at the beginning of the project, which studies ED users diagnosed with ACSC. In particular, it predates our study of persistent frequent ED users. Therefore, our databases included only patients diagnosed with ACSC (1 hospitalization or 2 physician visits related to an ACSC in the two year period before the index date), regardless of the reasons for their ED visits during their follow-up period in the present study. The aim of this study was thus to examine persistent frequent use in a cohort which was already established as ACSC diagnosed ED users. 

In the statistical analyses it is not clear the reason for splitting the sample into training and validation ones.

Response: The reason for splitting the cohort into training and validation samples was to avoid some statistical issues such as overfitting and to allow for estimations that are more robust. Moons et al. (2015) established guidelines for transparent reporting of a multivariable prediction model – part of the EQUATOR network – and recommended sample splitting as part of a robust design. Split sample validation is valid for prognostic model, which is why we used it only for the objective 2). It can be random or temporal; we chose the latter as it is considered a stronger approach in this context (intermediate between internal and external validation). It can lead to under or overestimation when there are strong temporal dependencies, however we also evaluated the occasional and persistent frequent ED use models 1) on the whole sample, 2) with a random split, and reached the same conclusions. We have added some details about the procedure (p. 11).

Also, the comparison of the results of only two groups (“persistent & frequent” vs. “frequent but not persistent”) seems insufficient. A better option will be to include also a third group of “no frequent users”.

Response: We thank you for this relevant suggestion. We updated the objectives and the results with the group “users other than persistent frequent users”, the complementary population to persistent frequent users (our primary interest).

Journal Requirements

Thank you for submitting the above manuscript to PLOS ONE. We note you have stated the following in the data availability statement:

"The datasets generated and/or analysed during the current study are not publicly available due to individual privacy. The CAI (Commission d'accès à l'information duQuébec) does not authorize data sharing outside of the research team."

Please note that PLOS journals require authors to make all data underlying the findings described in their manuscript fully available without restriction, with rare exception. PLOS journals will not consider manuscripts for which authors will not share data because of personal interests, such as patents, commercial interests or potential future publications. Your current statement in not in line with our data availability policy, which can be found at the following link: https://journals.plos.org/plosone/s/data-availability

Please update your Data Availability Statement to indicate whether you will be able to make your data available at the time of acceptance and provide details of where the underlying dataset can be found. Please be aware that without such confirmation we won't be able to consider your manuscript for publication. 

Please also note that we only require you to provide the minimal dataset, to consist of the data set used to reach the conclusions drawn in the manuscript with related metadata and methods, and any additional data required to replicate the reported study findings in their entirety. We do not require that you submit the entire data set if only a portion of the data were used in the reported study.

Response: We agree that data availability is an important issue in health research. We also agree that personal and commercial interests should not interfere with data availability. However, our research team is bound by legal reasons to not divulge any part of the data. The Commission de l’accès à l’information du Québec (CAI) is the provincial organism that allows researchers to access health databases. It is also responsible for ensuring their privacy as those databases contain sensitive patient information and it does not legally allow for making any part of them public. Therefore, we are not able to do so.

Reviewer #1

Thank you for the opportunity to read this manuscript, which concerns itself with a) estimating the prevalence of persistent frequent ED use, 2) identifying factors associated with this use and 3) comparing characteristics of frequent and non-frequent users. While I have offered comments to strengthen the paper should these authors resubmit to PLOS-One or elsewhere, I am not sure that it can make a contribution to this literature for reasons I detail in my comments. Ultimately, the fact that the study is described as a population based cohort study but the outcome variable was operationalized for only a subset of cases (and the fact that this is not clear in the manuscript) detract from the paper and preclude me from being able to recommend it for publication. I offer comments by section.

Response: We thank the reviewer for its comments; they helped us improve the manuscript. Please see below for our detailed answers.

Introduction. My initial comments were relative to the number of and appropriateness of the references in the introduction section. I would like to have seen more references (and potentially more explication) for the first set of assertions in the paper (e.g. the entire first paragraph). For example, even the second sentence (“Definition of frequent users varies depending on the context, though the most common definitions include having more than three or four visits during a 12-month period”) only includes one citation, the pertinence of which is not immediately appreciable. The premises in this paragraph form the bulk of their argument and feel insufficiently motivated (referenced) and under also under-nuanced. Is this true for cancer patients, patients with cancer related pain, the frail elderly, etc.? I am, in general, against language that suggests that patients “over-use” emergency care, when the situations of these patients’ lives is often such that it does represent the best care option for them at the time they choose it. Similarly, what is meant by ‘vulnerable’ patients on line 60? More detail and nuanced description of patterns of emergency care use are needed overall in the introduction. Although they have cited Billings (2013), it feels as through they have missed the subtleties of this piece.

Response: We agree that frequent ED use is a complex and diverse situation. Regarding the second sentence, we cited Krieg et al. (2016) because it is a recent scoping review, therefore a straightforward way to assess the definition of frequent ED user. We have added some references about the end of life and about patients with asthma, along with more details and nuances in the introduction (p. 4-5).

As another example, the exclusion from the cohort of patients who had died within 3 years of their index visit strengthened my opinion that the intro section could do a better job with the nuances of what constitutes ‘frequent and persistent use’ as in some cases (terminally ill patients) there may well be another definitions. I had also initially commented that looking deeper into this literature would also provide guidance for these investigators about whether it is in fact true that “no study exploring persistent frequent use with ACSC has been published.” However – as I read further in the manuscript and identified the issue of ‘outcome cases’ vs ‘reference cases’, and realized that the analyses in the paper are actually relative to ‘persistent and frequent users’ vs ‘frequent users’, I felt that the intro should also be revised in light of this focus. The reader assumes that the study is about ‘high users’ vs ‘non high users’ when in in fact is it about ‘persistent high users’ vs ‘episodic high users’. The literature review provided does little to shed light on hypothesis that may relate to ‘those with frequent use’ vs ‘those with persistent frequent use’. In fact, it is a bit misleading to imply that this is a ‘population based cohort study’ when in fact less than 10% of the sample is utilized for the primary analytic aim of the study.

Response: This is a population-based cohort study as the whole population was indeed included for the estimations. In particular, in the logistic regression models, the validation sample was used to estimate the odds ratios of being a “persistent frequent user” (versus “all other users”, therefore including the whole sample). We have split the population as a way to obtain estimations that are statistically robust (there are more details about the splitting strategy below, that have also been added to the text). The reviewer was right that the objectives 1) and 2) were about “persistent frequent users”, more precisely for 2) the case references for the odds ratios were “all other users other than persistent frequent users”. The objective 3) was about “persistent frequent users” versus “occasional frequent users” and “all users”. Regarding 3), a category “all others than persistent frequent users” has been added to the results. We added details in the methodology (p. 8-10).

Materials & Methods Section. Consistent with my comment above – I felt that the method section should make it clearer early on what cases comprise the actual analytic sample in this study. The reader does not know until Table 3. Figure 1 should be similarly revised to reflect this.

Response: This is a relevant comment on clarity. We have updated the text (p. 8-9) and Figure 1 (p. 7) accordingly.

Other comments: Is the RAMQ data source actually exhaustive? i.e. is it literally all adult patients in the province? Please detail, for American readers this is going to be a difficult concept to grasp. Does it contain all the variables used, or is it matched with (an)other data source? It wasn’t entirely clear to me if the RAMQ and the RAMQ administrative data are the same. If not, was it possible to match all cases in the cohort across the databases used? Cases lost (if any) during this process should be detailed in Fig. 1.

Response: The Régie de l’assurance maladie du Québec (RAMQ) is the provincial organism in charge of universal healthcare services; it therefore manages information about all residents living in the province of Québec. We used exhaustive health databases owned by the RAMQ in this study: 1) patient demographic register, which provides information on sex, date of birth, date of death and history of place of residence; 2) physician reimbursement claim register, which includes data on all medical services provided by a fee-for-service physician in Quebec: date of service, place of service (emergency, medical clinic, etc.), physician specialty, diagnosis (ICD-9), medical act procedure performed by the physician and the associated cost; and 3) the hospital register (MED-ECHO), which contains reasons for hospitalisation, that is, main diagnosis and up to 25 secondary diagnoses (ICD-10), dates of admission and release from hospital, and all medical procedures performed. We have added details in the text (p.6).

As cases were excluded for rural residential status with no/low metropolitan influence (but cases with rural status with high metropolitan influence were included), this variable should be more clearly defined.

Response: We agree with this comment. We thus have updated the text with the definition for remote areas and mentioned that the type of residential area was a categorical variable (p. 7-8).

The splitting of the testing and validation samples should be described in more detail, this is a somewhat unusual technique and deserves further explication. What is the rationale for this approach over others? Does it have any implications if there are temporal dependencies in the data?

Response: The reason for splitting the cohort into training and validation samples was to avoid some statistical issues such as overfitting and to allow for estimations that are more robust. Moons et al. (2015) established guidelines for transparent reporting of a multivariable prediction model – part of the EQUATOR network – and recommended sample splitting as part of a robust design. Split sample validation is valid for prognostic model, which is why we used it only for the objective 2). It can be random or temporal; we chose the latter as it is considered a stronger approach in this context (intermediate between internal and external validation), as mentioned in the same guidelines. It can lead to under or overestimation when there are strong temporal dependencies, however we also evaluated the occasional and persistent frequent ED use models 1) on the whole sample, 2) with a random split, and reached the same conclusions. We have added details about the procedure (p. 9-10).

Most importantly, and weaved throughout all my comments: In the section ‘Outcome and Dependent variables’, it should be made clearer that the binary outcome was ‘persistent & frequent users’ compared with the reference category ‘frequent but not persistent’. I was somewhat surprised when I reached the results section to see that this was the case. It is natural, given the language used and overall framing of this study, for the reader to assume that the reference category in analyses is going to be ‘non frequent (non persistent) user’ (e.g. the rest of the sample). This should also be addressed in figure 1, as the reader cannot tell (easily) from either the text or the figure what N was actually used in analyses (one does not see it until Table 3). It is difficult to tell if everyone who was not in the ‘persistent’ category were by default included in the other category? There must certainly have been cases for whom the index visit was the only visit? Is it fair to categorize them as ‘frequent but not persistent’? This is quite confusing until one reaches Table 3.

Response: Our objectives were 1) to estimate the prevalence of persistent frequent ED use; 2) to identify factors associated with persistent frequent ED use; and 3) to compare characteristics of persistent frequent users to occasional frequent users. Regarding objective 2), the evaluated category is “persistent frequent use” and the reference category is, as you suggested, “all others than persistent frequent use” (e.g. the rest of the sample, in which occasional frequent ED users and non frequent users are included). We realize it was not clear; we thus have updated the text (p. 5, 8 10) and Figure 1. Regarding objective 3), we wanted to evaluate any significant difference between occasional and persistent frequent users. This is a relevant comment and we updated objective 3) and results with “to compare characteristics of persistent frequent users to occasional frequent users and to users other than persistent frequent users” (p. 5, 13-15).

I was going to comment on the applicability of logistic regression with such a low frequency of cases on the outcome compared to the reference category, until I realized what was going on with the reference category (there is in fact a reasonable distribution of cases to controls in the analytic sample). Unless I am misunderstanding, these analyses really only include the approximately 13,000 cases categorized as either ‘persistent frequent’ or ‘frequent but not persistent’. This has to be made clearer throughout the manuscript. The rationale for not using the other (more obvious) reference category should be presented and as I said above, the introduction section should be made consistent with this approach. 

Response: The reviewer is right as in our regression analyses, the reference category for “persistent frequent users” was “all others than persistent frequent users”. We agree that this was not clear and, as mentioned in our previous response, we have updated the text accordingly. 

However, the paper still requires more detail on the backward selection technique used – did it result in only sex and age category as control variables, as is implied? This is confusing as Table 2 presents the odds ratios for many predictors in the multivariable model.

Response: The backward selection technique is an automated technique used for selecting independent variables in regression models, based on statistical criterion, and for developing parsimonious models. It starts from a full model (all independent variables included), then eliminates the non significant variable one at a time until all variables that remain are significant. We “forced” age and sex into the models as control variables, meaning we forced the selection process to keep them. We have updated the methods accordingly (p. 9-10).

Results. As the sample was split for training and validation it should be clarified in the Tables what sample(s) were used for which analyses. It is not stated what the N’s are that resulted from this process (size of each subsample), and as I highlight below, it remains somewhat unclear throughout the rest of the paper when the split samples were used, and what size the split samples were with respect to the total, frequent users, and persistent users. The fact that this study rests on approx. 13,000 cases is fine, but it should not be represented in the paper as being based on the full sample.

Response: We used the development sample for selecting the significant variables in the logistic regressions (n = 143,879), whereas we used the validation sample to estimate the odd ratios and the associated confidence intervals (n = 141,114). This information was clarified in Table 2, in Figure 1, and in the text (p. 9, 13-14).

Reviewer #2

The manuscript by Chiu YM et al. analyze in an adult population characterized by an ambulatory care sensitive conditions (ACSC) the prevalence of and the factors eventually associated to the condition of persistent frequent users in the Emergency Department. The manuscript is clearly written, data are well collected and the conclusions are well supported by novel and interesting results.

Response: We thank you for this positive feedback.

My main methodological concern is related to some criteria used for the selection of the experimental sample. In particular previous studies (see in particular PLOS ONE |2016 Dec 14;11(12):e0165939. doi: 10.1371/journal.pone.0165939. eCollection 2016.) have shown that older patients present clinical and social characteristics related to the definition of “elderly frail frequent users”. This data do not seem to be confirmed in the present study. Is it possible that the exclusion of patients affected by dementia and of patients died in the three years following their index may have caused a bias in the interpretation of data considering that dementia is strictly correlated with the old age? The authors need to discuss this issue.

Response: Our focus in this study was persistent frequent users (four ED visits per year over three consecutive years), which lead to us to exclude patients who died within three years of their index date. We also excluded patients who were diagnosed with dementia because of their specific characteristics; including them could have influenced our results generalization. However, those excluded patients may indeed have presented characteristics related to “elderly frail frequent users”. Our population being different from the one used by Legramante et al. (2016), this prevent us from drawing further conclusion. There are few papers on persistent frequent users, let alone on “elderly frail persistent frequent users” to compare results with those in the cited paper. However, we thank the reviewer for this comment and have added this issue to the limits (p. 19).

Furthermore the authors state that previous use of ED turned out to be the most important factor for all methods in encouraging the transition from occasional to persistent frequent users. My question is why does this happen? Which are the mechanisms linking the previous ED use to the transition in persistent frequent user?

Response: One of our main results is that the persistent frequent users included in our study were already frequent users in the year before the index date. Studies about frequent ED use also mention that previous ED use is the most important predictive variable. While we do not necessarily assume that this variable encourage the transition from occasional to persistent frequent use, we believe that persistent frequent users may have been holding that status for a longer period than studied here, as explained in p. 17-18 (we also added some details to the text). Because of the high rate of attrition of frequent users after the first year, most of them will not have as many ED visits as (already established) persistent frequent ED users, as seen in Table 3. Thus, this variable is of greater importance when it comes to studying persistent frequent use.

Can the authors provide some data about the hospital admission of the occasional and persistent frequent user? It is not clear to the reviewer the data “Previous hospitalization” in Tab. 3, please clarify this issue and eventually discuss these data.

Response: The variable “previous hospitalization” relates to having at least one hospitalization in the two years before the index date, regardless of the admission reason. Since we did not investigate reasons for hospitalization, we added this to the limits of the study.

Page 15, line 235: What do the authors mean for “...heavier ED history.”?

Response: Persistent frequent users had higher number of ED visits than frequent users (60 and 27 % had more than 4 ED visits respectively). We clarified this in the text (p. 16).

---

## [Decision Letter · Decision Letter 1]

15 Nov 2019

PONE-D-19-18906R1

Persistent frequent emergency department users with chronic ambulatory care sensitive conditions: a population-based cohort study

PLOS ONE

Dear Dr Chiu,

Thank you for submitting your manuscript to PLOS ONE. After careful consideration, we feel that it has merit but does not fully meet PLOS ONE’s publication criteria as it currently stands. Although the revised manuscript has improved, there are several important concerns that remain inadequately addressed.Therefore, we invite you to submit a revised version of the manuscript that addresses the points raised during the review process:

- Most of the mentions of ACSC found in the title, text and tables are misleading. Almost universally, ACSC are referred to preventable hospitalizations or ED visits potentially avoidable. The population of this study are all the patients with any of a group of chronic diseases that visited ER. Such diseases could be considered as ACSC if they would have been the reason for such visits. Consequently, in order to avoid confusion for the readers, the term ACSC should be replaced by “chronic conditions” or other similar.

- As reviewer 2 points out, I do not find a sufficient reason to exclude the patients with dementia. It seems plausible that such patients presented different health care needs to others subjects, but the same reason can be argued in relation to other diseases (for example HBP compared to asthma, COPD, or CHF) or age-groups.

- The authors should also follow the recommendations of reviewer 1 about the statistical model, inclusion of measures of model fit, and presentation of the results.

We would appreciate receiving your revised manuscript by Dec 30 2019 11:59PM. To enhance the reproducibility of your results, we recommend that if applicable you deposit your laboratory protocols in protocols.io, where a protocol can be assigned its own identifier (DOI) such that it can be cited independently in the future. For instructions see: http://journals.plos.org/plosone/s/submission-guidelines#loc-laboratory-protocols

We look forward to receiving your revised manuscript.

Kind regards,

Juan F. Orueta, MD, PhD

Academic Editor

PLOS ONE

Additional Editor Comments:

Besides, there are some minor errors in the draft. The description of the process to exclude patients presented in text (page 7) does not follow the same order as the one in figure 1; it seems to be a typo in the number and percentage of patients due to death. Also, in discussion (page 16, line 261) it is stated that “deprivation indices were not significant in our analyses”, in contradiction to the results of the study.

Reviewers' comments:

Reviewer's Responses to Questions

**Comments to the Author**

1. If the authors have adequately addressed your comments raised in a previous round of review and you feel that this manuscript is now acceptable for publication, you may indicate that here to bypass the “Comments to the Author” section, enter your conflict of interest statement in the “Confidential to Editor” section, and submit your "Accept" recommendation.

Reviewer #1: (No Response)

Reviewer #2: (No Response)

2. Is the manuscript technically sound, and do the data support the conclusions?

Reviewer #1: Yes

Reviewer #2: Partly

3. Has the statistical analysis been performed appropriately and rigorously? 

Reviewer #1: No

Reviewer #2: Yes

4. Have the authors made all data underlying the findings in their manuscript fully available?

Reviewer #1: No

Reviewer #2: No

5. Is the manuscript presented in an intelligible fashion and written in standard English?

Reviewer #1: Yes

Reviewer #2: Yes

6. Review Comments to the Author

Reviewer #1: I think the paper is improved with the changes. The purpose is clearer and the samples included in the analyses are also clearer. However, I continue to have some questions about the statistical approach and statistical conclusions, and the reporting of the results as well.

There is still an issue in these logistic regressions concerning the rarity of the outcome for the group ‘Persistent frequent users’, which comprise only 1/10 of 1% of the sample. As these models are inferential models intended to produce generalizable inferences about relationships between predictive factors and membership in a rare utilization group I think the models need more attention. Especially given the number of predictive variables (and their levels) considered in these models. (As an example, I wonder how many cases there are in the multidimensional cross-tabs of ‘age >-85 and ‘ has ACSC diabetes’, given all the other predictors. One might assume that this multidimensional cell size is quite small.) I think that in order for the results from these models to be credible some additional sensitivity analyses are warranted. I recommend considering Firths’ penalized logistic regression (Puhr R, Heinze G, Nold M, Lusa L, Geroldinger A. Firth‘s logistic regression with rare events – accurate effect estimates and predictions? Statistics in Medicine 2017.).

Additionally I strongly recommend the variable and model screening processes be reported as well as some measures of model fit, at a minimum an R2 measure, and possibly also ROC/AUC or information criteria measures for model fit and precision in order to allow readers to gauge for themselves how these models fit, and potentially how incrementally important the identified predictors are relative to a baseline model.

Reporting issues:

the statistical analysis section states that the first result is ‘prevalence of persistent frequent ED use’, however Table 1 does not contain that result (rather it reports ‘frequent users’). Is this a typo in the table?

Pg 11, line 207-208 should specify if the results presented in Table 2 represent the results of 2 separate models comparing 1) occasional frequent users to the entire sample and 2) comparing Persistent frequent users to the entire sample (vs comparing these two groups to each other) because it is still confusing. Additionally, the N’s should be reported in this table.

What are the cells in Table 2 that contain a ‘-‘? Were those coefficients not estimable?

Reviewer #2: The revised version of manuscript by Chiu YM et al. responded only partially to our concerns and also to the important concerns of other reviewers and to the Journal Requirements. In particular I continue to be convinced that not consider older people with chronic diseases such as dementia who, in my opinion, often could be considered as ambulatory care sensitive, may cause a bias in the full interpretation of the problem.

7. PLOS authors have the option to publish the peer review history of their article (what does this mean?). If published, this will include your full peer review and any attached files.

Reviewer #1: No

Reviewer #2: No

---

## [Author Response · Author response to Decision Letter 1]

16 Jan 2020

Dear Dr Orueta,

We would like to thank you again, along with the reviewers for their second round of comments on our manuscript. We believe that the comments allowed us to improve the quality of our manuscript. Please find attached the revised version and the detailed responses to each of the comments.

We hope the revisions will be to your satisfaction and we look forward to hearing back from you.

Best regards,

Yohann M. Chiu, Ph.D.

Corresponding Author

Université de Sherbrooke

3001, 12e Avenue Nord

Sherbrooke, QC J1H 5N4

Phone: 819-346-1110, #70538

Email: yohann.chiu@usherbrooke.ca

 

Editor’s comments 

Most of the mentions of ACSC found in the title, text and tables are misleading. Almost universally, ACSC are referred to preventable hospitalizations or ED visits potentially avoidable. The population of this study are all the patients with any of a group of chronic diseases that visited ER. Such diseases could be considered as ACSC if they would have been the reason for such visits. Consequently, in order to avoid confusion for the readers, the term ACSC should be replaced by “chronic conditions” or other similar.

Response: We agree; since we did not investigate the reason for ED visit, we have updated the text with “chronic conditions” instead of “ambulatory care sensitive conditions”.

As reviewer 2 points out, I do not find a sufficient reason to exclude the patients with dementia. It seems plausible that such patients presented different health care needs to others subjects, but the same reason can be argued in relation to other diseases (for example HBP compared to asthma, COPD, or CHF) or age-groups.

Response: We have run our analyses including this time patients with dementia, although this does not change the results or the interpretations. We have updated the manuscript with those latest results.

The authors should also follow the recommendations of reviewer 1 about the statistical model, inclusion of measures of model fit, and presentation of the results.

Response: We have updated the methods and the results according to the comments of the reviewer 1 (see below for detailed answers).

Besides, there are some minor errors in the draft. The description of the process to exclude patients presented in text (page 7) does not follow the same order as the one in figure 1; it seems to be a typo in the number and percentage of patients due to death. Also, in discussion (page 16, line 261) it is stated that “deprivation indices were not significant in our analyses”, in contradiction to the results of the study.

Response: The text for the description of the sample selection now follows Figure 1 (Lines 123-138); thank you for pointing out that. Regarding deprivation indices, it is true that they were significant in the occasional frequent use model, but they were not in the persistent frequent use model. We mentioned this difference in the discussion.

Reviewer #1

I think the paper is improved with the changes. The purpose is clearer and the samples included in the analyses are also clearer. However, I continue to have some questions about the statistical approach and statistical conclusions, and the reporting of the results as well.

There is still an issue in these logistic regressions concerning the rarity of the outcome for the group ‘Persistent frequent users’, which comprise only 1/10 of 1% of the sample. As these models are inferential models intended to produce generalizable inferences about relationships between predictive factors and membership in a rare utilization group I think the models need more attention. Especially given the number of predictive variables (and their levels) considered in these models. (As an example, I wonder how many cases there are in the multidimensional cross-tabs of ‘age >-85 and ‘has ACSC diabetes’, given all the other predictors. One might assume that this multidimensional cell size is quite small.) I think that in order for the results from these models to be credible some additional sensitivity analyses are warranted. I recommend considering Firths’ penalized logistic regression (Puhr R, Heinze G, Nold M, Lusa L, Geroldinger A. Firth‘s logistic regression with rare events – accurate effect estimates and predictions? Statistics in Medicine 2017.).

Response: We thank the reviewer for this pertinent suggestion. It is true that, given the small prevalence of persistent frequent use, specialized models should be used. We have run the analyses using Firth’s penalized logistic regression and have modified the methods and results sections accordingly. However, the results do not change in regards to numeric values or interpretations (though the updated results in the manuscript are slightly different as they now include patients with dementia, as suggested by another reviewer).

Additionally I strongly recommend the variable and model screening processes be reported as well as some measures of model fit, at a minimum an R2 measure, and possibly also ROC/AUC or information criteria measures for model fit and precision in order to allow readers to gauge for themselves how these models fit, and potentially how incrementally important the identified predictors are relative to a baseline model.

Response: Area under the curve, R2, and BIC have been added to the analysis (Table 2). However, since the goal is to evaluate explicative factors for persistent frequent use and not to compare different models, we did not compare our models to a baseline model. Line 229

Reporting issues:

The statistical analysis section states that the first result is ‘prevalence of persistent frequent ED use’, however Table 1 does not contain that result (rather it reports ‘frequent users’). Is this a typo in the table?

Response: We agree that it was not clear; we reported the prevalence of persistent frequent ED use in Table 1 but did not mention it. We clarified it in the table. Lines 207-208

Pg 11, line 207-208 should specify if the results presented in Table 2 represent the results of 2 separate models comparing 1) occasional Frequent users to the entire sample and 2) comparing Persistent frequent users to the entire sample (vs comparing these two groups to each other) because it is still confusing. Additionally, the N’s should be reported in this table.

Response: Table 2 indeed presents results for two regression models: 1) occasional frequent users versus the entire sample and 2) persistent frequent users versus the entire sample. We have clarified this in the text and added the sample sizes in the table. Lines 215-217

What are the cells in Table 2 that contain a ‘-‘? Were those coefficients not estimable?

Response: Table 2 contains results for two models: 1) occasional frequent use and 2) persistent frequent use. Since both models went through an automatic variable selection process, the explicative variables included were not necessarily the same for both; in particular, there were more variables for the first model. The dash ‘-‘ means that a variable was not selected therefore not estimated for the persistent frequent use model. We clarified this point in a footnote of Table 2. Line 229

Reviewer #2

The revised version of manuscript by Chiu YM et al. responded only partially to our concerns and also to the important concerns of other reviewers and to the Journal Requirements. In particular I continue to be convinced that not consider older people with chronic diseases such as dementia who, in my opinion, often could be considered as ambulatory care sensitive, may cause a bias in the full interpretation of the problem.

Response: Thank you for pointing out the possibility of a bias in the interpretations when excluding patients with dementia. We thus have run our analyses including them (n=12,189, 4.1 % of the total sample). Results and interpretations remain the same, though the material deprivation index is now significant in the logistic models. The sample selection, results and discussion are now updated. Lines 137; 155; 252

---

## [Decision Letter · Decision Letter 2]

29 Jan 2020

Persistent frequent emergency department users with chronic conditions: a population-based cohort study

PONE-D-19-18906R2

Dear Dr. Chiu,

We are pleased to inform you that your manuscript has been judged scientifically suitable for publication and will be formally accepted for publication once it complies with all outstanding technical requirements.

With kind regards,

Juan F. Orueta, MD, PhD

Academic Editor

PLOS ONE

Additional Editor Comments (optional):

Reviewers' comments:

Reviewer's Responses to Questions

**Comments to the Author**

1. If the authors have adequately addressed your comments raised in a previous round of review and you feel that this manuscript is now acceptable for publication, you may indicate that here to bypass the “Comments to the Author” section, enter your conflict of interest statement in the “Confidential to Editor” section, and submit your "Accept" recommendation.

Reviewer #1: All comments have been addressed

2. Is the manuscript technically sound, and do the data support the conclusions?

Reviewer #1: Yes

3. Has the statistical analysis been performed appropriately and rigorously? 

Reviewer #1: Yes

4. Have the authors made all data underlying the findings in their manuscript fully available?

Reviewer #1: Yes

5. Is the manuscript presented in an intelligible fashion and written in standard English?

Reviewer #1: Yes

6. Review Comments to the Author

Reviewer #1: (No Response)

7. PLOS authors have the option to publish the peer review history of their article (what does this mean?). If published, this will include your full peer review and any attached files.

Reviewer #1: Yes: Marianna LaNoue

---

## [Editor Report · Acceptance letter]

3 Feb 2020

PONE-D-19-18906R2 

Persistent frequent emergency department users with chronic conditions: a population‑based cohort study 

Dear Dr. Chiu:

I am pleased to inform you that your manuscript has been deemed suitable for publication in PLOS ONE. Congratulations! Your manuscript is now with our production department. 

With kind regards,

on behalf of

Dr. Juan F. Orueta 

Academic Editor

PLOS ONE